# Insights into the Anti-Adipogenic and Anti-Inflammatory Potentialities of Probiotics against Obesity

**DOI:** 10.3390/nu16091373

**Published:** 2024-04-30

**Authors:** A. K. M. Humayun Kober, Sudeb Saha, Mutamed Ayyash, Fu Namai, Keita Nishiyama, Kazutoyo Yoda, Julio Villena, Haruki Kitazawa

**Affiliations:** 1Department of Dairy and Poultry Science, Chittagong Veterinary and Animal Sciences University, Khulshi, Chittagong 4225, Bangladesh; humayuna2002@yahoo.com; 2Laboratory of Animal Food Function, Graduate School of Agricultural Science, Tohoku University, Sendai 980-8576, Japan; fu.namai.a3@tohoku.ac.jp (F.N.); keita.nishiyama.a6@tohoku.ac.jp (K.N.); 3Livestock Immunology Unit, International Education and Research Center for Food and Agricultural Immunology (CFAI), Tohoku University, Sendai 980-8576, Japan; 4Department of Dairy Science, Faculty of Veterinary, Animal and Biomedical Sciences, Sylhet Agricultural University, Sylhet 3100, Bangladesh; 5Department of Food Science, College of Agriculture and Veterinary Medicine, United Arab Emirates University, Al Ain 15551, United Arab Emirates; mutamed.ayyash@uaeu.ac.ae; 6Technical Research Laboratory, Takanashi Milk Products Co., Ltd., Yokohama 241-0023, Japan; k-yoda@takanashi-milk.co.jp; 7Laboratory of Immunobiotechnology, Reference Centre for Lactobacilli (CERELA-CONICET), Tucuman 4000, Argentina

**Keywords:** obesity, probiotics, anti-inflammatory adipokines, pro-inflammatory adipokines, adipocytes

## Abstract

Functional foods with probiotics are safe and effective dietary supplements to improve overweight and obesity. Thus, altering the intestinal microflora may be an effective approach for controlling or preventing obesity. This review aims to summarize the experimental method used to study probiotics and obesity, and recent advances in probiotics against obesity. In particular, we focused on studies (in vitro and in vivo) that used probiotics to treat obesity and its associated comorbidities. Several in vitro and in vivo (animal and human clinical) studies conducted with different bacterial species/strains have reported that probiotics promote anti-obesity effects by suppressing the differentiation of pre-adipocytes through immune cell activation, maintaining the Th1/Th2 cytokine balance, altering the intestinal microbiota composition, reducing the lipid profile, and regulating energy metabolism. Most studies on probiotics and obesity have shown that probiotics are responsible for a notable reduction in weight gain and body mass index. It also increases the levels of anti-inflammatory adipokines and decreases those of pro-inflammatory adipokines in the blood, which are responsible for the regulation of glucose and fatty acid breakdown. Furthermore, probiotics effectively increase insulin sensitivity and decrease systemic inflammation. Taken together, the intestinal microbiota profile found in overweight individuals can be modified by probiotic supplementation which can create a promising environment for weight loss along enhancing levels of adiponectin and decreasing leptin, tumor necrosis factor (TNF)-α, interleukin (IL)-6, monocyte chemotactic protein (MCP)-1, and transforming growth factor (TGF)-β on human health.

## 1. Introduction

The World Health Organization (WHO) states that obesity occurs when there is excessive fat accumulation in the body, which poses a risk to an individual’s health. A body mass index (BMI) greater than or equal to 25 (kg/m^2^) is considered overweight [1]. Obesity is a prevalent metabolic disorder characterized by excess white adipose tissue (AT) accumulation due to increased food intake and lifestyle changes [1,2]. Obesity is a medical problem that affects human health worldwide [1]. From 1980 to 2016, WHO states that there has been a nearly threefold increase in the number of obese individuals, marking it as a global epidemic. Globally, there is an increasing trend of obesity among all age groups. In 2016, an estimated 1.9 billion adults were overweight, of which approximately 650 million were living with obesity. Overall, ~13% of adults (11% men and 15% women) were obese in 2016 in the global population [1]. By 2030, the number of overweight and obese adults is predicted to reach 2.16 (38% of the global population) and 1.12 billion (20% of the global population), respectively [3].

Adipose tissue (AT) mass is determined by an increase in adipocyte size and the number of adipocytes, which comprise various cell types such as mature adipocytes, pre-adipocytes, fibroblasts, and macrophages [4]. The role of AT in obesity is crucial because inflammation increases the risk of obesity due to excessive accumulation of fat [5]. Functional foods containing probiotics provide beneficial nutritional enhancements for the treatment of overweight and related comorbidities [6,7]. Immunoregulatory probiotics have the potential to improve host intestinal conditions by modulating immune responses [8]. For instance, in vivo studies have demonstrated that probiotic administration lowers obesity caused by a high-fat diet, improves insulin resistance, and regulates the inflammatory response in mouse models [9] and human clinical trials [10]. Probiotics contain various microorganisms. The most common type of microorganism used as a probiotic is lactic acid bacteria (LAB), which have several health benefits through modulation of the immune system and enhancement of intestinal function [11]. To provide more basic research for probiotic application in obesity, porcine intramuscular preadipocyte (PIP) cell lines are useful in vitro tools to explore adipogenesis, AT immunobiology, and the therapeutic function of probiotics in obesity and its associated comorbidities [11].

Additionally, our laboratory and a few other in vitro studies have focused on the effects of probiotics on obesity [12,13,14,15] and the modulation of the immune response to prevent obesity, as demonstrated in this review [16,17]. Furthermore, the impact of probiotics overweight status has been confirmed by in vivo studies using animals and humans. Accumulating evidence suggests that different mechanisms link the effects of probiotics to obesity prevention and reduction. During obesity, a complex modification occurs in adipose tissue where numerous inflammatory markers and pro-inflammatory cytokines promote inflammation in adipose tissue [18]. However, the relationship between obesity and inflammation and the effect of probiotics on immune signaling in the obese state are not fully understood. Therefore, this review focuses on the role of probiotics, their effects, and the mechanisms involved in the regulation of immune pathways to prevent or treat obesity. Special interest has been focused on the most recent in vitro and/or in vivo (animal and human clinical) studies dealing with the use of probiotics to prevent obesity and its associated comorbidities. This review also addresses the limitations and future perspectives of probiotics in the treatment of obesity.

## 2. Methodology

A literature search was conducted in different electronic scientific databases (Google Scholar, Scopus, PubMed, Web of Science, Science Direct) using the keywords “obesity, probiotics, anti-inflammatory adipokines, pro-inflammatory adipokines, adipocytes”. The literature search considered published articles until December 2022 in search results, and repeated results were removed. An article search was conducted for articles published between August and December 2022. Five online databases were used to collect the articles. The databases used were Google Scholar (https://scholar.google.com/, accessed on 17 August 2022), Scopus (https://www.elsevier.com/ja-jp/solutions/scopus, accessed on 5 September 2022), PubMed (https://pubmed.ncbi.nlm.nih.gov/, accessed on 4 October 2022), Science Direct (https://www.sciencedirect.com/, accessed on 8 October 2022), and Web of Science (https://www.webofscience.com/wos/woscc/basic-search, accessed on 12 October 2022). Information for each source was collated considering the probiotics, experimental mode, animal, time, weight growth, free fatty acid/glycerides/cholesterol, anti-inflammatory adipokines, and pro-inflammatory adipokines. The documents were selected based on the criteria listed in Figure 1 according to Saha et al. (2023) [19].

## 3. Global Trends in Obesity

According to conservative estimates, 250 million people (approximately 7% of the global population) are obese worldwide [3]. Globally, obesity rates are likely to double by 2030, and approximately 573 million people will be obese [3]. Currently, approximately 130 million elderly people (over 20 years old) in the USA are overweight or obese. Overall, 31% of the men and 35% of the women older than 19 years are obese or overweight [3]. Obesity rates among children in the USA have significantly increased over the past three decades. About 20–25% of children are overweight or living with obesity, and the prevalence is higher in Pima Indians, Mexican Americans, and African Americans. According to Monitoring Trends and Determinants in Cardiovascular Disease, approximately 15% of men and 22% of women live with obesity in Europe [20]. In France, the number of obese individuals has increased from 8.5% in 1997 and 15% in 2012, with an increasing incidence of 0.43% per year [21]. For instance, the prevalence of obesity was 15% among 3–14-year-olds and 18.1% among 7–9-year-olds in France. The WHO European branch stated that the high prevalence of overweight individuals in Europe might soon be considered common [22]. The countries with the highest percentages of overweight boys and girls aged 11 years were Greece (33%), Portugal (32%), Ireland (30%), and Spain (30%), whereas the countries with the lowest percentages were the Netherlands (13%) and Switzerland (11%) [22]. The global distribution of obesity is shown in Figure 2. According to a 2007–2008 Chinese survey, 148.2 million persons had pre-diabetes, while approximately 92.4 million adults had diabetes because they were overweight [23]. Similar results have been reported in Asia and Africa. In the last 2–3 decades, obesity has been epidemic in Malaysia, Taiwan, China, Thailand, Australia, Pakistan, and Bangladesh [24]. Some countries in the Middle East, such as Saudi Arabia, Morocco, Bahrain, Tunisia, Egypt, Jordan, and Lebanon, have similar obesity rates often exceeding 40%. A recent study in the United Arab Emirates (UAE) reported that the total daily calorie intake is high in adolescents, resulting in nearly one-third (34%) of UAE children aged 12–18 years being overweight or obese [25]. In contrast, only 3.5% of Japanese people have a BMI > 30, which is based on the international standard for obesity [26]. According to the World Health Organization (2019) reports, obesity contributes to 44% diabetes, 23% ischemic heart disease, and 7–41% cancer burdens [27]. Moreover, obesity is linked to diabetes (80% related to obesity), hypertension (40% related to obesity), heart disease (70% linked to obesity), cancer (obesity accounts for 15–20% of cancer-related deaths), death (obese individuals have a 50–70% increased risk of death), and other metabolic problems. Obesity places a heavier burden on the health care system, leading to increased costs and a higher demand for services [28]. The annual cost of treating obesity in the USA is estimated to be approximately USD 100 billion annually [28].

## 4. Overview and Role of Adipose Tissue in Obesity

Adipose tissues or their precursor molecules are synthesized by de novo lipogenesis. The major function of these tissues is to regulate various biological process by secreting bioactive substances [29]. AT is a particularly active metabolic and endocrine organ that influences whole-body homeostasis by secreting substances such as growth factors, metabolites, hormones, and inflammatory cytokines. Obesity alters the secretion profile, causing negative effects on normal body physiology and consequent illnesses such as type II diabetes, hypertension, and chronic low-grade inflammation [30,31]. In general, AT is the major site for energy storage in the form of triglycerides/triglycerol (TG) in times of excess energy and is released as a source of energy, such as free fatty acids (FFAs) and glycerol, during periods of metabolic deprivation to maintain the body’s metabolic functions [32,33]. FFAs produced by lipolysis play an important role in the development of several metabolic diseases associated with obesity, including insulin resistance. FFAs can reach the liver directly through portal circulation in obesity, increasing the levels of hepatic FFAs responsible for improved lipid synthesis and gluconeogenesis, as well as insulin resistance in the liver [34]. In humans, elevated levels of circulating FFAs are believed to be responsible for peripheral insulin resistance [35].

Furthermore, FFAs act as endogenous ligands for the toll-like receptor 2 and 4 (TLR2 and TLR4) complex, causing macrophage cytokine release and regulating AT inflammation [36,37], which contributes to obesity-related metabolic syndrome. However, high FFAs concentrations in the blood do not always contribute to an increase in fat mass or indicate the onset of metabolic syndrome [38], although several recent studies have reported a link between FFA release from AT and obesity-related metabolic complications. Evidence shows that the adipose tissue acts as a primary endocrine organ, secreting a variety of adipokines including chemokines, cytokines, and hormones. Adipokines include various hormones, such as adiponectin, leptin, resistin, and visfatin, which play important roles in energy balance and inflammation, encompassing chemokines and cytokines [39]. Furthermore, adipocyte dysfunction contributes to the development of obesity-induced inflammation by enhancing the release of numerous pro-inflammatory chemokines and cytokines [34,39]. Monocyte chemotactic protein (MCP)-1, tumor necrosis factor (TNF)-α, interleukin (IL)-1, IL-6, and IL-8 have all been linked to insulin resistance [40,41]. The macrophage content of adipose tissue is positively linked to adipocyte size and body mass, and the appearance of pro-inflammatory cytokines such as TNF-α is mostly sourced from macrophages rather than adipocytes [42]. In summary, while recent results show that alterations in AT biology directly contribute to metabolic issues associated with obesity, the actual molecular factors connecting altered AT to peripheral organ diseases remain unknown.

## 5. Role of Probiotics in Human Physiology

In humans, food is mainly digested in the stomach and small intestine. The consumption of ultra-processed foods, which are usually rich in saturated and trans fatty acids, increases the risk of various diseases and induces dysbiosis. In addition, increased consumption of high-fat and high-sugar diets has been shown to change the gut microflora ecology, leading to a variety of illnesses [43]. Any disruption of the gut microbiota composition not only hampers host physiology but is also linked to several functions’ infections and disorders in the human body, such as bowel disease, antibiotic-induced diarrhea, diabetes, and obesity [44]. During this period, supplementation with specific probiotic strains can temporarily colonize the intestine and help to restore the disrupted gut microflora [45]. In doing so, probiotics help to re-establish essential physiological functions, promote commensal flora, and ultimately aid in mitigating physiological dysfunction in the human body.

## 6. Anti-Obesity and Anti-Inflammatory Effects of Probiotics

### 6.1. In Vivo Studies Performed in Animal Model

Several current studies have explored the potential benefits of probiotics for addressing obesity and inflammation in in vivo, with the findings detailed in Table 1. Most studies have reported that the anti-obesity effects of probiotics are mainly associated with the genus *Lactobacillus*, although some studies have shown that *Bifidobacterium* acts against obesity.

A previous study reported that the administration of *Lactobacillus curvatus* HY7601 or *Lactobacillus curvatus* HY7601 and *Lactiplantibacillus plantarum* KY1032 substantially decreased adipose tissue and body weight gain in mice fed a high-fat and cholesterol-rich diet for 9 weeks. The probiotic suppressed the plasma cholesterol (TG) level and reduced the fat accumulation and gene expressions of enzymes to synthesize fatty acid as well as pro-inflammatory cytokines (TNF-α and IL-1β) [46]. Similarly, another *Lactobacillus gasseri* strain, BNR17 (10^9^ or 10^10^ colony forming units (CFU)), was fed three types of diet (normal, high-sucrose, or high-sucrose diet containing *Lactobacillus gasseri* BNR17) for 10 weeks feeding of mice. Regardless of dosage, *Lactobacillus gasseri* BNR17 supplementation effectively reduced body weight gain, white adipose tissue weight, and total insulin, and leptin levels in the serum [47]. Furthermore, when *Bifidobacterium* spp. (*Bifidobacterium pseudocatenulatum* SPM 1204, *Bifidobacterium longum* SPM 1205, and *Bifidobacterium longum* SPM 1207) were added to high-fat diet-induced rats with obesity, they decreased the body and fat weights, blood serum parameters such as total cholesterol (TC), high-density lipoprotein *cholesterol* (*HDL-C*) low-density lipoprotein *cholesterol* (LDL-C), alanine aminotransferase (*ALT*)*,* TG, glucose, and lipase levels), and harmful enzyme activities (β-glucosidase, β-glucuronidase, and tryptophase) and efficiently augmented fecal LAB numbers [48].

Dietary supplementation with the probiotic strains *Lacticaseibacillus paracasei* CNCM I-4270, *Lacticaseibacillus rhamnosus* I-3690, or *Bifidobacterium animalis* subsp. *lactis* I-2494 reduced high-fat diet (HFD)-stimulated hyperglycemia, hyperinsulinemia, and glucose intolerance in HFD-fed mice. In addition to maintaining glucose–insulin homeostasis, these probiotic strains also reduced weight gain, hepatic steatosis, and macrophage infiltration into the epididymal adipose tissue [49]. Similar results were obtained in a study in which *Bifidobacterium adolescentis* was added to an HFD in rats [50]. Moreover, supplementation of *Lactococcus lactis* in a high-fructose diet for 42 days in mice resulted in diminished levels of hemoglobin A1c (HbA1c), fasting blood glucose, insulin, FFAs, and TG in the blood serum [51]. Another trial involving feeding high-fat diets for 8 weeks resulted in obesity in mice, after which they were randomly assigned to receive either an HFD along with *Lactobacillus curvatus* HY7601 and *Lactiplantibacillus plantarum* KY1032 or a placebo for an additional 10 weeks. Body weight gain was 38% lower in the probiotic-supplemented group than in the control group after the probiotic treatment. Surprisingly, the food conversion ratio and pro-inflammatory cytokines (TNF-α, IL-6, IL-1β, and MCP) decreased by 29% in mice treated with probiotics [52]. Additionally, supplementation with *Akkermansia muciniphila* showed anti-obesity properties, including reduced insulin resistance, adipose tissue inflammation, metabolic endotoxemia, and body weight gain [53]. Dietary supplementation with *Pediococcus pentosaceus* LP28/*Lactiplantibacillus plantarum* SN13T alongside high dietary fat resulted in enormous positive results in controlling obesity and reduction of FFAs in the blood. Administration of these probiotics reduced visceral fat accumulation, body weight gain, hepatic lipid content (TG and cholesterol), liver fat accumulation, and adipocyte size. Genes related to lipid metabolism that are affected by these factors include PPARγ, stearoyl-CoA desaturase-1 (SCD1), and cluster of differentiation 36 (CD36) [54]. Similarly, a 14-week study supplemented with *Bacteroides uniformis* CECT 7771 in the diet decreased epidermal adipose tissue, body weight gain, liver lipid contents (TG and cholesterol), plasma cholesterol, TG, glucose, insulin, and leptin levels and improved oral glucose in obese mice. *Bacteroides uniformis* CECT 7771 modulates the gut microbiota, increases the anti-inflammatory cytokines IL-10 and IL-33 in the intestine and peripheral blood, regulates T cells, and decreases the TLR5 expression in the ileum and epidermal adipose tissue [55].

Some probiotics have been explored for their potential immunomodulatory activities against obesity. *Lacticaseibacillus rhamnosus* administered to mice on a high-fat diet for 8 weeks resulted in lower duodenal IκB protein levels, along with the restoration of duodenal tight junction protein concentration. This strain was able to reduce portal LPS, TNF-α, IL-8, and IL-1β mRNA expression in the liver [56]. Similarly, db/db mice fed *Lacticaseibacillus rhamnosus* GG (LGG) for 4 weeks had a significant reduction in body weight gain, BMI, lipotoxicity, and decreased levels of pro-inflammatory adipokines (IL-6, TNF-α, and MCP1) [57]. A comparative study using four different *Bifidobacteria strains* (*Bifidobacteria* L66-5, L75-4, M13-4, and FS31-12) on lipid metabolism in HFD-induced obese mice revealed that all examined strains were beneficial for the reduction in TG in the liver and serum and substantially prevented fat deposition in the liver. The cholesterol content in the liver was markedly reduced by *Bifidobacterium* L66-5 and *Bifidobacterium* FS31-12 [58]. Additionally, probiotic *Bifidobacteria adolescentis* given to mice prevented liver damage, which was related to defense against lipid peroxidation, nuclear factor κB (NFκB) triggering, and liver inflammation. However, it did not have any effect on portal LPS, TLR-4, and myeloid differentiation primary response gene 88 (Myd-88) expression in the liver [59]. *Lacticaseibacillus rhamnosus* or *Bifidobacterium breve* and their combination have been demonstrated in studies on Zucker rats to lower serum lipopolysaccharides, which in turn prevents the development of hepatic steatosis and reduces TG in the liver. In addition to that mixture, all mentioned strains had the potential to reduce neutral lipids liver content, serum LPS levels, and serum TNF-α levels, and only *Lacticaseibacillus paracasei* reduced serum IL-6 levels [60].

Recently, a short-duration study of probiotic combinations containing 14 strains of *Lactobacillus*, *Lactococcus*, *Bifidobacterium*, *Propionibacterium*, and *Acetobacter* revealed that probiotic mixtures contribute to reducing body weight gain and visceral fat accumulation, hepatic lipid contents, serum cholesterol, TG, glucose, and levels of insulin and leptin, with subsequent improvement of insulin sensitivity and prevents the development of NAFLD in neonatal mice [61,62]. In addition, the strain *Lactiplantibacillus plantarum* FH185 strain supplementation reduced adipocyte size and gut microbial changes in diet-induced obese mice [63]. Another *Lactiplantibacillus plantarum* strain NTU102 and *Lactobacillus paracasei* NTU101 supplementation with soy milk inhibited adipocyte differentiation, suppressed *heparin-releasable lipoprotein lipase* activity, increased lipolysis activity, corrected dyslipidemia, and promoted serum leptin levels in HFD-fed rats [64]. A 12-week investigation was conducted in which mice were fed a low- or high-fat diet containing vancomycin, bacteriocin-producing *Ligilactobacillus salivarius* UCC118 (Bac+), or strain *Ligilactobacillus salivarius* UCC118 bacteriocin-negative (Bac-). Administration of vancomycin and bacteriocin-producing probiotics can markedly reduce weight gain and BMI, together with the level of pro-inflammatory adipokines (MCP1 and TNF-α) [65]. However, *Lactiplantibacillus plantarum* LMT1-48 helps to reduce obesity in HFD-fed mice by preventing the liver from making too much fat, causing a decrease in fat cells, and reducing belly fat and overall body weight [12]. Similar findings were observed with *Bifidobacterium longum* PI10, *Bifidobacterium animalis,* and *Lactobacillus gasseri* for 12 weeks, where the selected probiotics decreased body weight gain, fat deposition, obesity-linked metabolic dysfunction, and pro-inflammatory adipokines (IL-10 and leptin) [13].

In the animal studies, supplementation with probiotics reduced weight gain and cholesterol and improved adipokine profiles in obese individuals.

### 6.2. In Vivo Studies Performed in Human Clinical Trials

Recently, many human clinical trials have demonstrated that probiotics may be beneficial in the prevention and treatment of obesity (Table 2). For example, the effects of *Lactobacillus gasseri* SBT2055 (LG2055) were tested in Japanese adults with a large visceral fat area for 12 weeks. Participants were balanced and assigned randomly to three groups that received fermented milk (FM) containing LG2055 or without LG2055 and consumed 200 g/d of FM. The groups were instructed to consume 200 g FM daily for 12 weeks. The results showed that LG2055 reduced body weight and abdominal and visceral subcutaneous areas in adults [66]. Furthermore, it is advisable to drastically reduce the visceral and subcutaneous fat areas in the abdomen by an average of 4.6 and 3.3%, respectively. The beneficial effects of LG2055 on metabolism-related issues were demonstrated by a significant 1.5% reduction in body weight and BMI during treatment diet [66]. Another study by the same authors used the same strain and found that LG2055 reduced abdominal adiposity, body weight, other body measurements, and serum adiponectin levels in obese adults, indicating a beneficial effect on metabolic disorders [67]. In this study, 210 Japanese adults with large visceral fat areas were divided into three groups that received fermented milk (FM) containing 10^7^, 10^6^, or 0 (control) CFU LG2055/g of FM. Both LG2055 supplement groups showed a significant reduction in other measurements such as BMI, waist and hip circumferences, and body fat by Week 12; surprisingly, the consumption of FM for 4 weeks resulted in these beneficial effects. These observations indicate that supplementation with LG2055 at doses as low as 10^8^ CFU/day resulted in a significant reduction in visceral fat deposition, suggesting that constant consumption might be needed to maintain the effect. Dietary ingestion of specific probiotic bacteria decreases the inflammatory responses of aging and may have a dramatic effect on the management of obesity, without any side effects such as the drugs [67].

The results from an open-label study in patients with non-alcoholic fatty liver disease (NAFLD) by using “Symbiter” containing 14 alive probiotic bacteria of concentrated biomass showed a significant reduction in serum pro-inflammatory cytokines (IL-6, IL-10, and TNF-α) after 4 weeks of therapy in patients with normal or increased aminotransferase activity above the baseline [68]. Probiotics can be suggested for patients with NAFLD and type II diabetes, along with regular treatment, because they help to reduce low-grade systemic inflammation in the body [68]. Likewise, the administration of a tablet of 500 million *Lactobacillus bulgaricus* and *Streptococcus thermophilus* per day to patients with NAFLD for 3 months improved liver aminotransferase levels in NAFLD patients [69]. Another interesting study evaluated the efficacy of probiotics in patients with non-alcoholic steatohepatitis (NASH), with or without probiotic supplements, which had an impact on the levels of liver aminotransferases. After six months of treatment, probiotic supplementation (Protexin two tablets) along with metformin increased liver aminotransferase levels more than metformin alone in patients with NASH. By the end of the study, BMI and fasting blood glucose, cholesterol, and TG levels had significantly decreased [67]. Moreover, obese individuals show better metabolic syndrome with improved insulin sensitivity after translocation of the intestinal microbiota. At the end of 12 weeks, one study revealed that remarkable body weight changes were observed owing to alternating the metabolic functions [70].

A different open-label study that compared lifestyle modification with *Bifidobacterium longum* and fructo-oligosaccharides for 24 weeks in patients with NASH found significant reductions in TNF-α, CRP, serum aspartate transaminase (AST) levels, HOMA-IR, and serum endotoxin [71]. Similarly, a 6-month randomized open-label study comparing the lepicol probiotic formula (*Lactiplantibacillus plantarum*, *Lactobacillus delbrueckii*, *Lactobacillus acidophilus*, *Lacticaseibacillus rhamnosus*, and *Bifidobacterium bifidum*) with usual care revealed a statistically significant change in intrahepatic TG content compared with the usual care group. Conversely, no positive correlation was found between the use of probiotics and changes in BMI, waist circumference, and glucose or lipid levels. However, probiotic supplements may help patients with NASH to decrease liver fat and serum AST levels [72]. Mazloom et al. (2013) found that a 6-week oral administration of a combination of Lactobacillus probiotics, including *Lactobacillus acidophilus*, *Lactobacillus bulgaricus*, *Lactobacillus bifidus*, and *Lacticaseibacillus casei,* led to a reduction in TG levels, malondialdehyde (MDA) concentration, anti-inflammatory cytokines (IL-6) levels, and insulin resistance in patients with type II diabetes. However, these changes were not statistically significant [73]. Another probiotic mixture (*Lactobacillus acidophilus* La5, *Bifidobacterium lactis* Bb-12, and *Lacticaseibacillus casei* DN001) containing yogurt was supplied to people with high BMI who were randomly assigned into three groups depending on particular intervention diets: the initial group consumed regular yogurt as part of their low-calorie diet, the second group consumed probiotic yogurt with a low-calorie diet, and the third group consumed probiotic yogurt without a low-calorie diet (PWLCD) for two months [74,75,76]. The groups who received the probiotic yogurt with a low-calorie diet had lower BMI, fat, and leptin levels than the other diets but TNF-α and TGF-β gene expression were comparable. However, there was a decrease in IFN-γ expression in all treatment groups. These findings suggest that the gene expression profile in peripheral blood mononuclear cells among obese and overweight individuals was influenced by a low-calorie diet combined with probiotic yogurt [74,75,76].

The probiotic *Lactobacillus gasseri* BNR17 strain was supplied to obese adults daily before meals for 12 weeks and its effects on obesity were observed [77]. The study found no discernible differences were observed between the placebo and *Lactobacillus gasseri* BNR17 groups in terms of sex, age, medical treatment, medication use, smoking, drinking, or other personal habits. Nevertheless, the addition of *Lactobacillus gasseri* BNR17 is linked to a slight decrease in weight, waist circumference, and hip circumference, and may offer a novel and secure approach to weight management. Despite the limitations of this study, such as the brief trial period and unevaluated indices, it also implies that the investigated probiotics could effectively decrease body weight in obese or overweight conditions [77].

Several strains of *Lactobacillus* have been tested in human clinical trials, for example, the administration of *Lacticaseibacillus rhamnosus* GG probiotics to the mother for 4 weeks prior to delivery and for the first 6 months after a randomized, double-blind, prospective follow-up study [78]. The study demonstrated that the trial had an impact on the development of children and resulted in them becoming obese by the age of 10 years. Two phases of excessive weight gain were identified: the first phase began during the fetal period and continued until the child reached 24–48 months of age, and the second phase began after the child reached 24–48 months of age. The impact of the intervention was also demonstrated by a trend toward a reduction in the mean 4-year BMI adjusted for birth weight. Regulation of the gut microbiota with probiotics can help to prevent young children from gaining excessive weight during their first years of life [78]. Another probiotic, *Lacticaseibacillus paracasei* F19, was administered to 58 obese postmenopausal women for 6 weeks with flaxseed mucilage or a placebo. Compared to the placebo, feeding *L. paracasei* F19 did not alter any of the metabolic markers (HOMA-IR, Matsuda index, C-reactive protein (CRP), or lipid profile) [79]. Furthermore, *L. acidophilus* La5 and *B. animalis* subsp. *lactis* Bb12 supplementation did not affect HOMA-IR, BP, heart rate, or serum lipid content in individuals with increased weight [80,81].

Probiotic supplementation in the capsule form of *Bifidobacterium*, *Lactobacillus*, and *S. thermophilus* in overweight adults improved lipid profiles and HDL-C levels, while dramatically lowering TC, TG, and LDL-C levels. Moreover, the probiotic mixture enhanced insulin sensitivity, lowered CRP levels, and favorably modified the gut microbiota composition [82]. The administration of probiotics (*Lacticaseibacillus casei* Shirota, LcS) in conjunction with milk to patients with metabolic syndrome resulted in a reduction in vascular cell adhesion molecule-1. However, no significant changes were observed in the parameters associated with low-grade inflammation [83]. Insulin resistance in postmenopausal women is an important risk factor for cardiovascular morbidity, particularly stroke, coronary heart disease, and mortality. Over 90 days, the effectiveness of *Lactiplantibacillus plantarum* was assessed in postmenopausal women with metabolic syndrome by supplementing them with milk fermented with *Lactiplantibacillus* and non-fermented milk without *Lactiplantibacillus plantarum*. At the end of the study, the levels of TC, IL-6, and glutamyl transpeptidase (GTP) were significantly lower in both supplemented groups, whereas LDL-C was lower in the non-fermented group. Compared with the non-fermented group, the *Lactiplantibacillus plantarum*-containing fermented group showed reduced blood glucose and homocysteine levels [84].

In a recent study, 12 weeks of oral treatment with probiotic capsules containing *Lactiplantibacillus* and *Bifidobacterium breve* reduced waist circumference, total fat area, visceral fat, and leptin levels [85]. *Lactiplantibacillus* K50 administration for 12 weeks resulted in a reduced in cholesterol and triglyceride levels by altering the gut microbiota, illustrating the potential benefits of probiotics in controlling blood lipid profiles [86]. Overall, the use of probiotics may be a prospective approach for treating obesity and its associated comorbidities. Choosing probiotics for managing and treating obesity and its associated comorbidities is safe, as these have beneficial therapeutic effects reported as therapeutic means. Altering the gut microorganisms with probiotic supplements can influence body weight by controlling glucose and fat metabolism, enhancing insulin sensitivity, and reducing systemic inflammation. Next-generation probiotics have evolved for the successful recovery and prevention of these disorders.

### 6.3. In Vitro Studies Performed Using Different Animal Cell Lines

Several in vitro studies have been conducted using different cell lines such as PIP and 3T3-L1 cells, to screen for the anti-obesity effects of different probiotic strains. The PIP cell line was developed from Duroc pig Musculus longissimus thoracis, and the 3T3-L1 cell line was developed from mouse embryos. Both models are relevant for studying obesity and its associated comorbidities owing to their exposure to adipogenesis, and present similarities (anatomy and physiology) between pigs, mice, and humans. The in vitro PIP model is one of the most commonly used models to explore adipogenesis, AT immunobiology, and the curative role of probiotics in obesity and its associated comorbidities [16,17,87]. Several studies have focused on the in vitro evaluation of probiotic strains using different animal cell lines (Table 3). A recent in vitro PIP model study for assessing the immunomodulatory and anti-adipogenic activities of LAB showed that mature adipocytes from PIP cells treated with cell-free supernatant (CFS) from *Lacticaseibacillus rhamnosus* LA-2 and *Lacticaseibacillus casei* TMC0409 strains had significantly lower lipid deposition than the control groups. Also, PIP cells pretreated with CFS from *Lacticaseibacillus rhamnosus* GG, *Lacticaseibacillus paracasei* TMC0409, or *Bifidobacterium bifidum* TMC3108 exhibited enhanced expression of TGF-β at Hour 6 post-TNF- treatment, but at Hour 12, those cells showed lower levels of IL-6 and MCP-1. *Lacticaseibacillus rhamnosus* LA-2 was able to dramatically reduce IL-6 and MCP-1 levels but did not affect TGF-expression. This study indicates that *Lacticaseibacillus rhamnosus GG, Lactobacillus gasseri* TMC0356, and *Lacticaseibacillus rhamanosus* LA-2 significantly inhibited pro-inflammatory cytokines and chemokines expression in adipocytes subjected to TNF-α [16]. Thus, the same research group investigated to investigate whether innate immune ligands have any effect on the adipogenesis of the PIP cell line and CCL2, IL-6, IL-8, and IL-10 expression [16]. These findings suggest that innate immune ligands have distinct effects on adipogenesis in the PIP cell line, particularly with respect to differences in poly (I:C) and LPS-induced fat accumulation. Furthermore, there is a direct link between the TLR pathway in adipocytes and the integration of innate immunity with adipocyte functions [17]. Activation of TLRs in mature porcine mature adipocytes is responsible for transcript modification which is linked to the immune response and cellular lipid metabolism [87].

A co-culture model was used to investigate the interactions between intestinal immunocompetent cells and probiotics in obesity. In the co-culture model, adipocytes and the probiotic mixture were used as the probiotic control group. *Bifidobacterium bifidum* TMC3108 was selected for its anti-obesity properties and capacity to modulate the immune system. A strain-dependent effect was observed when analyzing fat accumulation and CCL2 protein production in adipocytes in the co-culture model. The probiotic strains *Lacticaseibacillus rhamnosus* GG, *Lacticaseibacillus rhamnosus* LA-2, and *Lacticaseibacillus paracasei* TMC0409 were not able to increase the expression of the adipokine CCL2, while *B. bifidum* TMC3108 increased CCL2 production in adipocytes. However, *Lacticaseibacillus rhamnosus* GG, *Lacticaseibacillus rhamnosus* LA-2, and *Lacticaseibacillus paracasei* TMC0409 were found to modulate inflammation and reduce fat accumulation, together with immunocompetent cells and probiotics. Remarkably, anti-obesity capacity was observed in *Lacticaseibacillus rhamnosus* LA-2 and *Lacticaseibacillus paracasei* TMC0409 [16,17]. Overall, these findings suggested that LAB (certain strains with immune regulatory effects) strains could decrease PIP differentiation via immune cell activation and Th1 cytokine production [16].

Another probiotic combination with heat-killed *Lacticaseibacillus rhamnosus* GG and *Lactobacillus gasseri* TMC0356, used in the murine macrophage-like cell line J774.1 cells and later transferred to the preadipocyte cell line 3T3-L1, resulted in PPAR-γ mRNA expression and enhanced IL-6 and IL-12 production reported in J774.1 cells treated with GG and TMC0356. These findings suggest that lactobacilli modify macrophage activation-mediated suppression of preadipocyte differentiation and alter macrophage immune responses to adipose cells [14]. Another in vitro study on 3T3-L1 adipocyte cell culture with kimchi (a traditional Korean fermented vegetable) supplemented with *Leuconostoc mesenteroides* KCCM 11353P and *Lactiplantibacillus plantarum* KCCM11352P showed a significant reduction in the number of lipid droplets and lysis of globular lipid droplets within 24 h of treatment [15]. In addition, lower levels of TG and higher levels of intracellular glycerol in the probiotic-treated group than in the control group indicated that lipolysis was enhanced by probiotics, thus playing an important role in obesity reduction [15]. In addition, treatment with the *Lactiplantibacillus plantarum* strain LMT1-48 downregulated the lipogenic genes, as evidenced by the inhibition of 3T3-L1 adipocyte differentiation and lipid accumulation [12]. Several other immunoregulatory probiotic strains (*Lacticaseibacillus rhamnosus* LA-2, *Lacticaseibacillus casei* TMC0409) have demonstrated the ability to modulate fat accumulation and adipokine secretion in PIP cells [17]. For example, a recent in vitro study on 3T3-L1adipocyte cell cultures treated with a mixture of *Bifidobacterium longum* PI10, *Bifidobacterium animalis* LA804, *and Lactobacillus gasseri* LA806 showed a significant reduction in lipid accumulation within 84 h of treatment [13].

In summary, many studies (in vivo and in vitro) have been conducted using various probiotic strains, dosages, and administration times to evaluate the effects of probiotics on obesity. These data suggested that most probiotic strains exhibited positive effects against obesity.

## 7. Limitations of Using Probiotics in Different Research Models (In Vitro and In Vivo)

In vitro studies have some limitations that must be considered when using different cell lines (3T3-L1, PIP, etc.), because every cell line has different features. Variations in microenvironmental parameters from the optimal level of experimental conditions may influence the expression of the examined traits. Molecular and genomics-based studies in vivo will aid in the identification of genuine probiotics and selection of the most appropriate ones to reduce obesity.

Nevertheless, more studies are needed for the following: (i) to determine whether the probiotic interventions in early life stages help to decrease the risk of obesity; (ii) to test whether small changes in energy utilization by different microbiota markedly affect weight gain and obesity in humans; (iii) to test the results of several studies which have shown significant differences in the gut microbiota populations between people living with obesity and those living in lean conditions and, whether these differences lead to obesity or obesity itself leads to differences; (iv) to test the role of ‘novel’ beneficial microbes as therapeutic tools; (v) to obtain conclusive proof of the preventative and curative effects of probiotics in medical practice, well-designed clinical studies including a large number of patients from ethnically homogeneous groups, are required; (vi) to provide more human clinical studies and meta-analyses necessary to prove the efficacy of traditional probiotics for treating obesity; and (vii) to determine the dietary and genetic factors related to obesity. Future research on probiotics should concentrate on understanding their function in vivo under various obesity-related conditions and on increasing their acceptance by medical professionals. To obtain the greatest benefits, doctors, nutritionists, and scientists must work closely and share information with healthcare professionals and the public.

## 8. Mechanisms of Action of Probiotics as Anti-Obesity

Molecular mechanisms and specific pathways by which probiotics prevent, control, and fight obesity are unclear, but numerous mechanisms have been proposed (Figure 3). Probiotics have various modes of action, including enhancing beneficial microbiota, inhibiting the growth of pathogens, preventing the binding or penetration of pathogenic bacteria into the intestinal mucosal layer, enhancing mucosal barrier activities, producing antimicrobial chemicals, changing immunoregulation, reducing pro-inflammatory molecules, and stimulating preventive molecules [49].

LAB may have the potential to reduce body weight [62]. Oral intervention with LAB stimulates anti-obesity potential by altering the intestinal microbiota composition, regulating energy metabolism and absorption of nutrients, and maintaining immunity [88,89,90]. Gut microbiota can fight obesity by modifying the energy balance and metabolic functions of the host [91]. Different types of microbiota have different roles in the body; some obesity potentiates, while others play a role against obesity [92]. Number, type, and microbiome composition are the most important factors to consider when choosing medication for obesity. Most LAB fight obesity through various pathways. LAB are widely used as probiotics in the production of ferment foods. First, the available probiotics are made from one species but are now common mixed species of *Lactobacillus* and *Bifidobacteria.* These probiotics directly affect the intestinal flora and influence other organs through immunomodulation, increasing intestinal permeability and producing regulatory or bioactive molecules [93,94]. Immune responses may be activated by immunomodulatory components of LAB such as peptidoglycans and bacteriocins [13,86]. Furthermore, *Lactobacillus* can reduce the expression of various pro-inflammatory cytokines, chemokines, and TLRs in macrophages during the feeding of enterohemorrhagic *E. coli*-infected mice. LAB-containing probiotics can improve intestinal barrier function, thereby preventing intestinal infection and inflammation. *Bifidobacterium lactis* Bb12 exhibits an excellent capacity to adhere mucus, more than double that of the GG strain of *Lactocaseibacillus* or *Lactobacillus delbrueckii* subsp. *bulgaricus,* by preventing the adsorption and colonization of pathogens on intestinal epithelial cells [95]. By enhancing the barrier function, LAB may reduce the release of LPS from intestinal epithelial cells and the production of pro-inflammatory cytokines in adipose tissues [96].

Probiotics combat obesity through various pathways. Multiple species of probiotics alter gut microbiota populations, thus changing the short chain fatty acids (SCFAs)-hormone pathway and bile acid synthesis, resulting in a reduction in fat accumulation, suppressing weight gain, de novo cholesterol synthesis, and insulin resistance [97]. Furthermore, some probiotic strains modulate energy and nutrient metabolism. *Lacticaseibacillus rhamnosus* PL60 produces conjugated linoleic acid, which has the potential to reduce body fat reserves, thereby reducing obesity without decreasing energy intake [98]. These researchers also suggested another possible mechanism, as the anti-obesity effects of probiotics are related to apoptosis and messenger RNA expression in white adipose tissue. *Lacticaseibacillus rhamnosus* PL60 supplementation increased signals of apoptosis, and uncoupling protein levels in adipose tissue, and serum leptin, which can contribute to reducing obesity. The administration of *Ligilactobacillus salivarius* UCC118Bac+ (producing bacteriocin) or the bacteriocin-negative strain *L. salivarius* UCC118Bac− (non-bacteriocin-producing) to rats can modulate the gut microbiota [99]. However, only the bacteriocin-producing probiotics increased the number of Bacteroidetes and Proteobacteria and decreased the number of Actinobacteria compared with the non-bacteriocin-producing control treatments. Researchers have confirmed that long-term probiotic feeding may be required to manipulate the gut microbiota [100,101]. Our research group revealed that *Lactobacilli* (some strains with immunoregulatory effects) may suppress differentiation of pre-adipocytes through immune cell (dendritic cells) activation and creation of Th1 cytokines comprising IFN-γ, IL-2, and TNF-α which may reduce obesity [102]. Specifically, some strains of *Lactobacillus* and *Bifidobacterium* are known to maintain the Th1/Th2 balance and contribute to prevention of adipocyte hypertrophy. *Lactiplantibacillus plantarum* CBT LP3 and *Bifidobacterium breve* CBT can regulate the Th1/Th2 balance and reduce obesity symptoms [85].

Various probiotics alter mucosal barrier permeability, thereby suppressing excess lipids and enhancing monosaccharide absorption. Oral administration of *Bifidobacterium breve* through fermented food reduces the absorption of excess lipids [101]. An important mode of action of probiotics is the modulation of energy metabolic hormones and factors. Supplementation of *Bifidobacterium breve* B-3 in a HFD to mice reduced epidermal fat accumulation and body weight by increasing the expression of genes associated with fat metabolism and insulin sensitivity [103]. Similarly, *Lactiplantibacillus plantarum* (LP14) also decreases the mean size of adipocytes, weight of white adipose tissue, and serum levels of TC [104]. Another probiotic strain, *Lactobacillus gasseri* BNR17, can reduce obesity by upregulating the expression of genes related to fatty acid oxidation-related genes [47]. Probiotics containing *Streptococcus thermophilus* increase the release of SCFAs which stimulate the release of GLP-1 from intestinal cells, resulting in reduced food intake and improved glucose tolerance [105].

Obesity is linked to the chronic activation of inflammatory pathways in AT [106]. Chronic low-grade inflammation (systemic or metabolic) is an important initial event in the development of obesity induced complications. Notably, insulin resistance has been linked to chronic AT inflammation. A common severe inflammatory response includes inducer molecules (e.g., lipopolysaccharide (LPS)), known as endotoxins which are recognized by pathogen recognition receptors (e.g., TLR4) which trigger the production of inflammatory mediators [e.g., TNF-α and IL-6] that cause an inflammatory response in target cells (e.g., endothelial cells and adipocytes) [107]. Furthermore, LPS produces reactive oxygen species via a respiratory burst as part of the host immune response to pathogens. Disruption of the gut microbiota composition results in elevated levels of LPS in the blood or liver, provoking hepatic inflammation, such as NAFLD. In patients with NAFLD, increased levels of prooxidative NADPH oxidase 2 (NOX2) activity have been observed as a result of elevated serum sp-NOX2, which leads to systemic oxidative stress [108]. Another marker, 8-iso-PGF2 alpha, is high in the urine and indicates increased oxidative damage in NAFLD [109]. Oxidative stress impairs insulin signaling and promotes fat accumulation in the body and inflammation in the liver. In general, when the number of beneficial bacteria decreases, the number of harmful bacteria that can produce LPS in the gut increases. LPS can then stimulate TLR4 and the downstream signal via My88, and the TRIF pathway activates mediators to trigger inflammation in adipocytes [110]. Inflammation involves a carefully managed process that facilitates the transition to the homeostatic state. Adenosine monophosphate kinase (AMPK) is an important skeletal muscle enzyme that contributes to maintaining cellular homeostasis. TLR4 expression is related to AMPK activity, which is a key regulator of fatty acid metabolism [111]. The presence of numerous harmful bacteria in the gut suppresses AMPK, resulting in obesity [112].

Obesity is related to chronic low-grade inflammation caused by circulating concentrations of cytokines and acute-phase proteins. The activation of tissue-specific genes that promote inflammation in particular areas of the body has been described, leading to an increase in macrophages within enlarged tissues [67,68,69]. Hypertrophied adipocytes and local hypoxia in an enlarging AT can induce endoplasmic reticulum stress via oxygen reactive stress within fat cells, which leads to the activation of inflammatory pathways in macrophages and adipocytes. As a result, more macrophages infiltrated the AT, causing local and systemic inflammation and insulin resistance (Figure 3). In summary, evidence from in vitro and in vivo investigations indicates that obesity-associated macrophage infiltration into AT is responsible for both local and systemic inflammation and that probiotic intake protects against obesity through various mechanisms. It seems that the primary mechanism of action is related to changes in the intestinal microbial composition. Supplementation with probiotics can alter the intestinal microbiota by increasing *Lactobacilli* and *Bifidobacterium* spp. However, further research is needed to clarify how probiotic bacteria modulate the gut microbiota and contribute to preventing obesity by regulating various pathways.

## 9. Conclusions

The prevalence of obesity is increasing worldwide and presents a significant and alarming trend in many countries. This surge in obesity rate is linked to a higher likelihood of premature death. Based on current scientific evidence derived from in vitro, animal, and human clinical research, probiotics have emerged as an effective and targeted approach for combating obesity. Furthermore, it appears that probiotics have master-switch roles in gut microbiota composition and function against obesity (a significant reduction in body weight gain, abdominal adiposity, BMI, pro-inflammatory adipokines and an improved serum lipid profile, anti-inflammatory adipokines, metabolism of carbohydrates, etc.) and its associated comorbidities, such as insulin resistance, type II diabetes, and non-alcoholic fatty liver disease. In individuals with non-alcoholic fatty liver disease, probiotics predominantly improved liver function and metabolic issues. Considering these data, the use of probiotics to control or prevent obesity has gained considerable interest. Probiotics are a promising approach for combating obesity. However, the limitations of vitro and/or animal and human clinical studies include small sample sizes, short-term studies, and the absence of long-term follow-up studies. Therefore, further clinical trials are required to validate the advantageous impact of probiotics against obesity and the quest for next-generation probiotics to treat obesity and its associated comorbidities.

## Figures and Tables

**Figure 1 nutrients-16-01373-f001:**
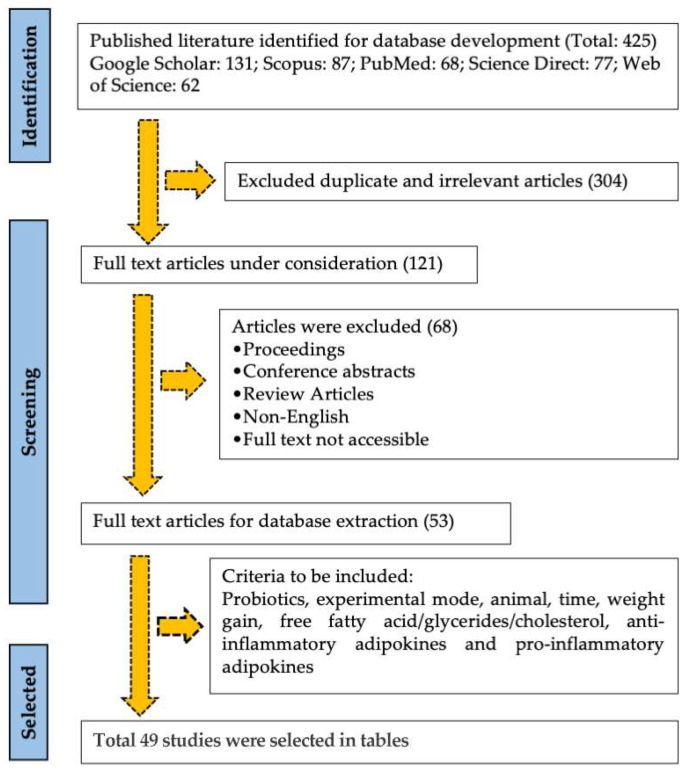
Flowchart of scientific literature search and selection for study by following PRISMA guidelines.

**Figure 2 nutrients-16-01373-f002:**
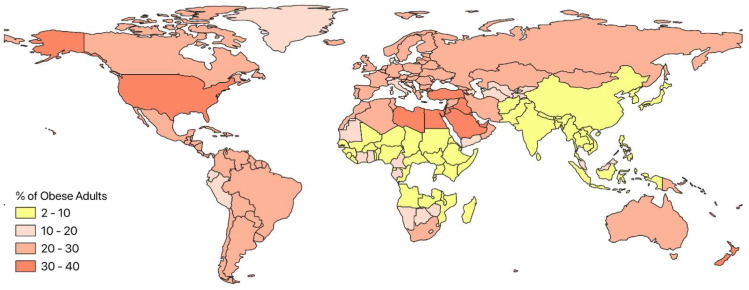
Global obesity rate distribution map (Source: NCD ris C estimates, https://ncdrisc.org/data-downloads-adiposity.html, accessed on 14 June 2023).

**Figure 3 nutrients-16-01373-f003:**
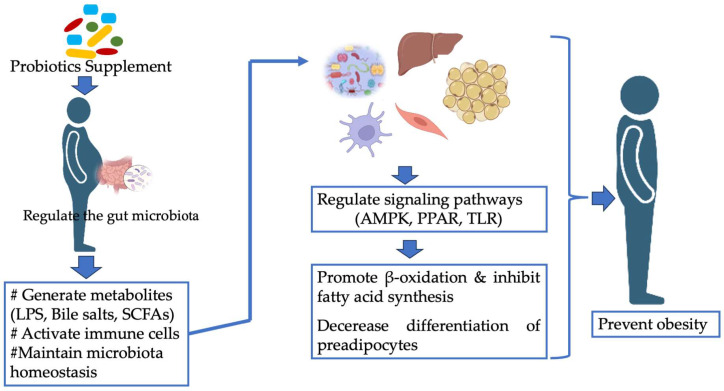
Probiotics’ mechanisms of anti-obesity effects. Abbreviations: LPS, lipopolysaccharides; SCFAs, short-chain fatty acids; AMPK, adenosine monophosphate-activated protein kinase; PPAR, peroxisome proliferator-activated receptor; TLR, toll-like receptors.

**Table 1 nutrients-16-01373-t001:** Summary of current animal model (in vivo) for evaluating the effects of probiotics as anti-obesity and anti-inflammatory activities.

Probiotics Species/Strains	Expt. Mode	Animal	Time	Effects with Respect to				Ref.
				Weight Gain	FFA/TG/Chole-Sterol	Anti-Inflammatory Adipokines	Proinflammatory Adipokines	
*Lactiplantibacillus plantarum* LMT1-48	HFD-induced	Mice	5 wk	↓	↓	-	-	[12]
*Bifidobacterium longum* PI10, *Bifidobacterium animalis* and *Lactobacillus gasseri* LA806	HFD-induced	Mice	12 wk	↓	↓	-	↓ IL-10 and Leptin	[13]
*Lactobacillus curvatus* HY7601or along with *Lactiplantibacillus plantarum* KY1032	High-fat high-cholesterol diet	Mice	9 wk	↓	↓	-	↓TNF-α and IL-1β	[46]
*Lactobacillus gasseri* BNR17	High-sucrose diet/standard *chow-fed*	Mice	10 wk	↓	↓	-	↓ Leptin	[47]
*Bifidobacterium pseudocatenulatum* SPM 1204, *Bifidobacterium longum* SPM 1205, and *Bifidobacterium longum* SPM	High dietary fat induced rat	Rat	7 wk	↓	↓	-	-	[48]
*Lacticaseibacillus rhamnosus* I-3690, *Lacticaseibacillus paracasei* CNCM I-4270, or *Bifidobacterium lactis* I-2494	HFD-induced/standard *chow-fed*	Mice	12 wk	↓	-	-	↓ TNF-α	[49]
*Bifidobacterium adolescentis*	HFD-induced/standard *chow-fed*	Rat	12 wk	↓	↓	-	-	[50]
*Lactococcus lactis*	Diet containing high fructose/standard *chow-fed*	Rat	42 d	-	↓	-	-	[51]
*Lactobacillus curvatus* HY7601 and *Lactiplantibacillus plantarum* KY1032	HFD-induced/standard *chow-fed*	Mice	10 wk	↓	↓	-	↓ TNF-α, IL-6, IL-1β and MCP-1	[52]
*Akkermansia muciniphila*	HFD-induced/standard *chow-fed*	Mice	4 wk	↓	-	-	-	[53]
*Pediococcus pentosaceus* LP28/*Lactiplantibacillus plantarum* SN13T	HFD-induced/standard *chow-fed*	Mice	6 wk	↓	↓	-	-	[54]
*Bacteroides uniformis* CECT 7771	HFD-induced/standard *diet*	Mice	14 wk	↓	↓	↑ IL-10, IL-33, thymic stromal lymphopoietin	TSLP	[55]
*Lacticaseibacillus rhamnosus* GG	High-fructose diet	Mice	8 wk	↓	↓	-	↓ TNF-α, IL-8 and IL-1β	[56]
*Lacticaseibacillus rhamnosus* GG	Induced diabetic	Mice	4 wk	↓	↓ Lipotoxicity	-	↔ TNF-α and IL-6 ↓ MCP-1	[57]
*Bifidobacteria* L66-5, L75-4, M13-4 and FS31-12	HFD-induced/standard *chow-fed*	Mice	6 wk	↓ B. L66-5 ↑ B.M13-4 ↔L75-4 and FS31-12	↓	-	-	[58]
*Bifidobacterium adolescentis*	HFD-induced/standard *chow-fed*	Mice	12 wk	↓	-	-	↑ CCL2	[59]
*Lacticaseibacillus paracasei* CNCMI-4034, *Bifidobacterium breve* CNCM I-4035 and *Lacticaseibacillus* *rhamnosus* CNCM	Genetically obese animal	Rat	30 d	↓	↓	↓ Adiponectin	↓ TNF-α and IL-6	[60]
14 probiotics of *Bifidobacterium*, *Lactobacillus*, *Lactococcus*, *Propionibacterium* genera	Obesity created with Monosodium glutamate	Rat	3 mon	↓	↓	↓ Adiponectin	↔ Leptin	[61]
14 probiotics of *Bifidobacterium*, *Lactobacillus*, *Lactococcus*, *Propionibacterium* genera	Monosodium glutamate induced obesity	Rat	3 mon	↓	↓	-	-	[62]
*Lactiplantibacillus plantarum* FH185	High dietary fat induced mice	Mice	6 wk	-	↓	-	-	[63]
*Lacticaseibacillus paracasei* subsp. *paracasei* NTU 101	HFD-induced/standard *chow-fed*	Rat	5 wk	↓	↓	-	↓ Leptin	[64]
*Ligilactobacillus salivarius* UCC118	A low fat (lean) or diet-induced obese	Mice	8 wk	↓	↓	-	↓TNF-α and MCP1	[65]

Abbreviations: ↑, increased; ↓, decreased; ↔, no significant difference between groups; -, not assessed; FFA, free fatty acid, TG, triacylglycerols; IL, interleukin; MCP-1, monocyte chemotactic protein; CCL2, chemokine ligand 2; TNF-α, tumor necrosis factor-α.; IFN-γ, interferon-γ; wk, week; mon, month; d, day.

**Table 2 nutrients-16-01373-t002:** Summary of current human clinical trials measuring effects of probiotics in obesity and its associated comorbidities in humans.

Probiotics Species/Strains	Type of Study/Subjects	Time	Effects with respect to					Ref.
			Weight Gain	FFA/TG /Cholesterol	Anti-Inflammatory Adipokines	Pro-Inflammatory Adipokines	Other Measures	
*Lactobacillus gasseri* SBT2055 (LG2055)	A multicenter, double-blind, randomized, placebo-controlled intervention trial	12 wk	↓	↓	↑ Adiponectin	-	↓ Abdominal adiposity	[66]
*Lactobacillus gasseri* SBT2055 (LG2055)	Multicenter, double-blind, parallel-group randomized controlled trial (RCT)	12 wk	↓	↓	-	-	↔ Lean mass and subcutaneous fat area, ↓ BMI	[67]
Multiprobiotic “Symbiter” containing concentrated biomass of 14 alive probiotic bacteria	Open-label study in patients with non-alcoholic fatty liver disease	4 wk	-	-	-	↓ IL-6, IL-8, and TNF-α	↓ Low-grade systemic inflammation	[68]
*Lactobacillus bulgaricus* vs. *Streptococcus thermophilus*	Randomized, double-blind, parallel, placebo-controlled trial	3 mon	-	-	-	-	↓ Liver aminotransferases levels in patients with NAFLD	[69]
Probiotic combination with Metformin 500 mg (Met/Pro) versus Metformin 500 mg plus placebo (Met/P)	Double-blind, randomized, placebo-controlled trial in patients with histology-proven—NASH	6 mon	↓ BMI	↓	-	-	↓ Liver aminotransferases level and fasting blood glucose, ↑ Liver function	[70]
*Bifidobacterium longum* with fructo-oligosaccharides	Open-label study in patients with NASH	24 wk	↓ BMI	↓	-	↓TNF-α	↓ CRP, serum AST, HOMA-IR, serum endotoxin, steatosis, NASH activity index	[71]
Lepicol probiotic formula *(Lactiplantibacillus plantarum, Lactobacillus deslbrueckii, Lactobacillus acidophilus, Lacticaseibacillus rhamnosus* and *Bifidobacterium bifidum)* vs. usual care	Randomized, open-label study in patients with histology-proven NASH	6 mon	↔ BMI	↔	-	-	↓ AST	[72]
Lactobacillus probiotics *(Lactobacillus acidophilus, Lactobacillus bulgaricus, Lactobacillus bifidus* and *Lacticaseibacillus casei)*	Randomized, single-blinded, placebo-controlled trial	6 wk	-	↔	-	↔ IL-6	↓ IR	[73]
*Lactobacillus acidophilus* La5, *Bifidobacterium* Bb12 and *Lacticaseibacillus casei* DN001	Subjects with high BMI	8 wk	↓ BMI	↓	↓ Leptin	↔ TNF-α and TGF-β ↓ IFN-γ	Changes in gene expression in PBMCs	[74,75,76]
*Lactobacillus gasseri* BNR17 in capsule	Double-blind RCT placebo group	12 wk	↓	↔	-	-	↔ BMI, fat, waist circumference, VAT, SAT, DAT, ↑ HDL, ↓ LDL	[77]
*Lacticaseibacillus rhamnosus* GG, ATCC 53103	Randomized, double-blind, prospective follow-up study	Mothers: 4 wk ≥ EDD Child: ≤6 mon	↓	-	-	-	↓ Weight gain (1st yr)	[78]
*Lacticaseibacillus paracasei* N19	Obese post menopausal women	6 wk	-	-	-	-	↑ Insulin sensitivity, ↑ Gut microbiota	[79]
*Lactobacillus acidophilus La5* and *Bifidobacterium animalis subsp. Lactis* Bb12	Overweight adults	6 wk	↔ BMI	↔	-	-	↓ Fasting glucose, ↑ HOMA-IR	[80,81]
*Bifidobacteria, lactobacilli,* and *Streptococcus thermophilus*	Overweight subjects	6 wk	-	↓	-	-	↑ Insulin sensitivity, ↓ CRP, ↑ HDL	[82]
*Lacticaseibacillus casei* Shirota	Patients with IRS	12 wk	↑	-	-	↔ TNF-α and IL-6	↓ VCAM-1	[83]
*Lactiplantibacillus plantarum*	PM women with IRS	12 wk	↔	↓	-	↓ IL-6	↓ Glucose, homocysteine	[84]
*Bifidobacterium breve* CBT and *Lactiplantibacillus plantarum* CBT LP3	Randomized, double-blind, placebo-controlled trial	12 wk	↔	↓	-		↓ HDL, glucose, and insulin	[85]
*Lactiplantibacillus plantarum* K50 (LPK)	Randomized, double-blind, placebo-controlled trial	12 wk	↔	↓	↓ Leptin	-	-	[86]

Abbreviations: ↑, increased; ↓, decreased; ↔, no significant difference between groups; -, not assessed; FFA, free fatty acid, TG, triacylglycerols; BMI, body mass index; CRP, C-reactive protein; HOMA-IR, homeostasis model assessment of insulin resistance; IR, insulin resistance; IRS, insulin resistance syndrome; PBMC, peripheral blood mononuclear cell; PM, postmenopausal; NASH, non-alcoholic steato-hepatitis; AST, aspartate transaminase; VAT, Visceral adipose tissue; DAT, deep adipose tissue, HDL, high-density lipoprotein, LDL, low-density lipoprotein; VCAM-1, vascular cell adhesion molecule-1; wk, week; mon, month; yr, year.

**Table 3 nutrients-16-01373-t003:** Summary of application of probiotics in cell culturing model (in vitro) for measuring the effects of probiotics as anti-obesity and anti-inflammatory activities.

Probiotics Species/Strains	Expt. Mode	Animal	Time	Effects in Respect to				Ref.
				Weight Gain	FFA/TG/Cholesterol	Anti-Inflammatory Adipokines	Pro-Inflammatory Adipokines	
*Lactiplantibacillus plantarum* LMT1-48	In vitro (3T3-L1 model)	Mice	48 h	-	↓	-	-	[12]
*Bifidobacterium longum* PI10, *Bifidobacterium animalis,* and *Lactobacillus gasseri* LA806	In vitro (3T3-L1 model)	Mouse	48 h	-	↓	-	↓ Leptin	[13]
*Lacticaseibacillus rhamnosus GG* and *Lactobacillus gasseri* TMC0356	In vitro (3T3-L1 model)	Mice	24 h	-	↓	-	↑ IL-6 and IL-12	[14]
*Leuconostoc mesenteroides* and *Lactiplantibacillus plantarum*	In vitro (3T3 L1 model)	Mice	3 wk	-	↓ Adipogenesis/Lipogenesis ↑ Lipolysis	-	-	[15]
*Lacticaseibacillus rhamnosus* GG, *Lactobacillus gasseri* TMC0356, and *Lacticaseibacillus rhamnosus* LA-2	In vitro (porcine intramuscular preadipocyte; PIP model)	Pig	48 h	-	↓	-	↑ IL-6, MCP-1, and TGF-β	[16]
*Bifidobacterium bifidum* TMC3108; LGG, *Lacticaseibacillus rhamnosus* GG; LA-2, *Lacticaseibacillus rhamnosus* LA-2; TMC0409, *Lacticaseibacillus paracasei* TMC0409	In vitro (PIP model)	Pig	48 h	-	↓	↑ CCL2	↓ IFN-γ, IL-6, 12	[17]

Abbreviations: ↑, increased; ↓, decreased; -, not assessed; IL, interleukin; MCP-1, monocyte chemotactic protein; CCL2, chemokine ligand; TGF-β, transforming growth factor-β; IFN-γ, interferon-γ, wk, week; h, hour.

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
