# Peer review of "Insights into the Anti-Adipogenic and Anti-Inflammatory Potentialities of Probiotics against Obesity"

_nutrients, 2024, doi:10.3390/nu16091373_

Round 1

Reviewer 1 Report

Comments and Suggestions for Authors

Should be of interest to analyze the behaviour of circulaing endotoxing  and discuss the role of oxidant stress, previously related to microbiota in NAFLD.

So he Authors should analyze markers of oxidant stress, such as serum sp-NOX2 and urinary 8-iso-PGF2 alpha.

Comments on the Quality of English Language

Moderate editing of English language is required.

Author Response

Dear Reviewer,

We are very grateful for your comments on the manuscript. According to your
advice, we have revised the manuscript. Our responses to your comments appear in the attached file.

Thank you!!

Best regards

Sudeb Saha, PhD

Reviewer 2 Report

Comments and Suggestions for Authors

The review could be refined by focusing on the following general points:

The first half of the title is fine not sure what the other half pertains to?

Obesity is defined as a disease by some. How the development of obesity can be retarded as opposed to how reduction/cure in pre-existing obesity (cure, a term used by the authors) could easily be different.

There are sections on the global trend in obesity, role of adipose tissue in obesity. Since the focus is on the link between probiotics and obesity, it is important that there is a section on Probiotics and their role in human physiology and any epidemiological evidence linking use of probiotic – food with chronic diseases, focusing on obesity.

This review assesses the role of probiotics in the development and/or reduction in obesity. The current review provides evidence from studies conducted in pre-clinical models as well as intervention trials conducted in population-based studies.

Preclinical models especially those in animal studies combines obesogenic diet with probiotics and asses the effect of weight gain and various markers of inflammation (obesity prevention) whereas human intervention trials assess the effect on preexisting obesity on weight reduction and pro-inflammatory markers etc. (reduction and treatment of obesity).

A clear distinction must be made between prevention and treatment.

The reviewer searched PubMed for articles related to this review. Several articles and their contents will add value to the current review and can be included in this review. The reviewer admits that some articles may have been included.

Selected articles are listed below:

• Effects of Probiotics and Synbiotics on Obesity, Insulin Resistance Syndrome, Type 2 Diabetes and Non-Alcoholic Fatty Liver Disease: A Review of Human Clinical Trials

Maria Jose Sáez-Lara, Candido Robles-Sanchez, Francisco Javier Ruiz-Ojeda, Julio PlazaDiaz, Angel Gil Int J Mol Sci. 2016 Jun; 17(6): 928. Published online 2016 Jun 13. doi: 10.3390/ijms17060928

• Emerging issues in probiotic safety: 2023 perspectives

Daniel Merenstein, Bruno Pot, Gregory Leyer, Arthur C. Ouwehand, Geoffrey A. Preidis, Christopher A. Elkins, Colin Hill, Zachery T. Lewis, Andi L. Shane, Niv Zmora, Mariya I. Petrova, Maria Carmen Collado, Lorenzo Morelli, Gina A. Montoya, Hania Szajewska, Daniel J. Tancredi, Mary Ellen Sanders Gut Microbes. 2023; 15(1): 2185034. Published online 2023 Mar 15. doi: 10.1080/19490976.2023.2185034

• Probiotics: How Effective Are They in the Fight against Obesity?

Kiran Mazloom, Imran Siddiqi, Mihai Covasa Nutrients. 2019 Feb; 11(2): 258. Published online 2019 Jan 24. doi: 10.3390/nu11020258

• Effects and Mechanisms of Probiotics, Prebiotics, Synbiotics, and Postbiotics on Metabolic Diseases Targeting Gut Microbiota: A Narrative Review

Hang-Yu Li, Dan-Dan Zhou, Ren-You Gan, Si-Yu Huang, Cai-Ning Zhao, Ao Shang, Xiao-Yu Xu, Hua-Bin Li Nutrients. 2021 Sep; 13(9): 3211. Published online 2021 Sep 15. doi: 0.3390/nu13093211

• The Role of Probiotics and Prebiotics in the Prevention and Treatment of Obesity

Tomás Cerdó, José Antonio García-Santos, Mercedes G. Bermúdez, Cristina Campoy Nutrients. 2019 Mar; 11(3): 635. Published online 2019 Mar 15. doi: 10.3390/nu11030635

Comments on the Quality of English Language

Minor editorial changes are required.

Author Response

Dear Reviewer,

Thank you for your precise time in reviewing our paper and providing valuable comments. Please find the attached file for the point by point responses of comments.

Best regards

Sudeb Saha, PhD

Reviewer 3 Report

Comments and Suggestions for Authors

In this review, Humayun Kober et al. discussed the potential of probiotics in combating obesity, highlighting their anti-adipogenic and anti-inflammatory effects. The authors exhaustively summarize various studies (in vitro and in vivo) on how probiotics reduce body weight, improve insulin sensitivity, and regulate immune responses to prevent obesity. The topic of this review is interesting, and the manuscript is well-written and engaging. Table 1-3 summarized by the authors provide a clear view of the role of probiotics in anti-obesity. Here are some comments on this study:

1.       Abstract, line 27 “microflora”, it is recommended that authors use the “microbiota”, which would be more consistent throughout the manuscript. It is recommended that authors define abbreviations of TNF-α, IL-6, MCP-1, and TGF-β.

2.       Lines 136-137 “In the last 2-3 decades, obesity was found in epidemic form in Malaysia, Australia, New Zealand, Taiwan and China [24]”, I could not find any information about Malaysia New Zealand, and Taiwan in reference 24.

3. The title of section 4 is “Overview and role of adipose tissue in obesity”, it is suggested that the authors revise the title to “Overview and role of probiotics in obesity”.

4.       Line 404 “first years of life74”, a different format.

5.       Section 6, since the authors mention the role of metabolites (line 574 SCFAs) of gut microbiota in obesity, it is inevitable to mention bile acids, which play a crucial role in fat absorption.

Author Response

Dear Reviewer,

Thank you for your valuable time to review our manuscript. We have tried our best to address evry one of the comments. We hope that manuscript after careful revisions meet your high standard.

Best regards

Sudeb Saha, PhD

Round 2

Reviewer 1 Report

Comments and Suggestions for Authors

The Authors answered correctly to all my queries

Comments on the Quality of English Language

Moderate editing of English language is required